# Proper Technical Maintenance of Combine Harvester Rolling Bearings for Smooth and Continuous Performance for Grain Crop Agrotechnical Requirements

**Eglė Jotautienė, Antanas Juostas and Shankar Bhandari \***

Institute of Agricultural Engineering and Safety, Agriculture Academy, Vytautas Magnus University, LT-53362 Akademija, Lithuania; egle.jotautiene@vdu.lt (E.J.); antanas.juostas@vdu.lt (A.J.)
**\*** Correspondence: shankar.bhandari@vdu.lt

**Abstract:** The threshing mechanism is the main component of the combine harvester on which the grain separation and cleaning qualitative work indicators depend. It is important to ensure that all threshing mechanism components, including the threshing drum bearings and all other bearings of the combine, are working properly and reliably. There are many places in the combine where it is not possible to measure bearing vibrations directly without dismounting them, since there is no suitable spot to mount a sensor. The paper investigates the threshing drum rolling bearing condition of combines, which are difficult to access, by using a vibration diagnostics technique utilizing a newly manufactured steel bracket. The vibration measurements and analysis were conducted by the Adash A4900 Vibrio M analyzer (Adash spol. s.r.o., Ostrava, Czech Republic). The vibration source measurement was based on the fast Fourier transform (FFT) spectrum analysis. Analysis of the experimental results showed that average squared velocity values (in the frequency interval of 10–1000 Hz), together with other measured vibration parameters, can be used for the combine threshing drum's bearing condition evaluation.

**Keywords:** combine harvester; threshing drum; diagnostic; vibrations; rolling bearing





## 1. Introduction

Combine harvesters operate in varying crop and field conditions, such as under conditions of high moisture, environment dustiness, and exposure to straw material. The harvesting season is typically short. During harvesting time, crops might be wet, laid down, and vary in straw content. In addition, due to field conditions, the driving speed of the combine varies as well. These conditions could lead to a variable amount of feeding crop to the threshing mechanism of the harvesting combine, resulting in an excessive load on the combine drive mechanisms and their bearings. Such an excessive load not only aggravates the threshing process, but also harms gears and decreases their longevity. Very small soil particles rise into the air during the movement of combine harvesters, tractors, and other agricultural machinery. This leads to the accumulation of soil dust on bearing housings, and eventually penetrates the bearings. In addition, the bearing units experience significant dynamic load in the form of shocks and vibrations [1].

The threshing mechanism is the key part of the combine harvester on which the grain separation and quantitative cleaning work indicators depend. Thus, it is important to ensure that all the components of the threshing mechanism, including the threshing drum bearings and all other bearings of the combine, are properly functioning.

Proper technical maintenance ensures smooth and continuous performance that fulfills the agrotechnical requirements with the least amount of crop losses and expense. The rolling element of the bearing is the basic component of almost all mechanisms that transmit power in agriculture equipment, aircraft, ships, vehicles, various industrial installations, etc. [2].

Unexpected damage to the bearings can lead to disruption in the operation of equipment and financial expense. Damaged bearings often destabilize the rotation of the shaft, and can result in disruption to the operation of the entire mechanism. To avoid such incidents, it is important to identify any bearing defects as early as possible. The diagnostics of bearing defects have been investigated frequently in the past few years [3]. Bearing defect diagnostics include the time domain, frequency domain, combined time–frequency analysis (as presented in [4,5]), vibration–acoustic measurements, temperature measurements, and wear debris analysis. Vibration measurement diagnostics seem to be the most widely used [6].

Vibration signature diagnostics allow the condition of operating machinery to be checked. They can identify potentially approaching failures at a particular location, avoiding significant damage to the machinery that could lead to expensive repairs. Periodic scheduled vibration measurements could ensure the reliable operation of machinery [7]. Vibration analysis provides pertinent information regarding the progressive deterioration of bearings, and gives a baseline signature for future monitoring. Frequency domain spectrum analysis could identify defect frequencies, and predict the presence of defects in roller bearings [8]. In addition, multiple faults can be detected using vibration analysis in the frequency domain mode [9].

Vibration analysis techniques for rolling bearing fault diagnostics are extensively used by researchers. When the bearing rollers are operating in a damaged cage, vibration shock impulses are generated. However, similar vibration signatures can also be produced by various other mechanical units and mechanisms, such as worn-out gears [6,10–13]. Bearing defects are recognized not only by measuring the velocity; if the periodic vibration measurements indicate that the bearing velocities are increasing, it could mean that bearing defects are serious, and it could lead to a serious mechanical failure. However, a measurement of the bearing rotational speed does not provide the required information of bearing status. The bearing vibration velocities need to be measured to assess the bearing condition. Numerous researchers have found that time domain analysis is able to indicate the presence of bearing faults, but it cannot identify the location of these faults. Frequency domain analysis can identify the location of bearing faults. In the measured response spectrum, response peaks occur at certain bearing fault frequencies, from which the location of defects in the bearing mechanism can be identified. In addition, envelope analysis can also be a very useful method to detect incipient defects of the rolling bearing. Combined time–frequency domain analysis could detect bearing faults effectively due to its very fine response data resolution [14]. Wavelet analysis is also a good method for fault diagnostics. A significant amount of research has been undertaken on artificial neural networks (ANNs) and fuzzy logic for bearing fault diagnostics [15]. These techniques can detect bearing faults very quickly, and with high accuracy.

A static and modal analysis of the engine mounting bracket was performed to support the engine, which sustains the vibration caused by the engine and tires due to bumping on the road surface. The natural frequency was identified from finite element analysis (ANSYS) for the gray cast iron bracket, with which its low natural frequency and mode shapes by modal analysis proved to be restrictive in the vibration characteristics of the bracket [16]. A static and modal analysis of the supporting L-shaped bracket for air the compressor system was performed to find its natural frequency and mode shapes. The modal analysis avoids the resonance phenomenon by proper selection of materials and design to avoid the common natural frequencies in the system, in order to eradicate the problems of inefficiency of the system and financial loss [17]. In this paper, modal analysis and harmonic response analysis were also performed on the bracket in ANSYS software, to identify and eradicate the common natural frequencies which could result in resonance.

Known bearing vibration testing methods are more often adopted and developed for industrial machines than for agricultural ones. Therefore, we needed to consider how to use the existing bearing testing methodologies for mobile agricultural machines. There are cases where it is not possible to directly measure bearing vibrations due to a lack of access

to a required measurement location. This is the case for the harvester threshing drum bearing. The objective of the current research was to develop a vibration measurement procedure that could provide the required information to assess rolling bearing defects. The main novelty of this study was the use of a specialized bracket and its implementation for a bearing test in a difficult-to-access component of mobile agricultural machines. By implementing the bracket, the testing equipment was able to be used to measure the bearing vibrations of agriculture machines, without dismounting the bearing, as after dismounting the threshing drum bearings, reinstallation of the old bearings is not feasible.

## 2. Materials and Methods

A straw walker combine mechanism was chosen for the present research project. The investigated threshing drum bearings were type 20213-K-TVP-C3 (FAG, Schweinnfurt, Germany). All bearing vibration measurements were repeated three times at three bearing locations. The measurements were carried out on three new and one used combine with an output of 3590 MH, at several threshing drum speeds. The same type of bearings of these combine threshing drums are used in various Lexion combines.

The Adash A4900 Vibrio M (Adash spol. s.r.o., Ostrava, Czech Republic) vibration accelerometer with a magnetic base and the Adash A4900 Vibrio M vibration analyzer were used for vibration measurement and data analysis. This type of equipment is useful in detecting early warnings related to bearing conditions for industrial applications, such as high-speed turbines, rotors, gearboxes, etc. This instrument satisfies the ISO 10816-3 standard [18] and the vibration testing requirements specified in ISO 10816-3, 2009 ISO 10816-3:2009 mechanical vibration—evaluation of machine vibration by measurements on non-rotating parts, Part 3: industrial machines, with nominal power above 15 kW and nominal speeds between 120 min$^{-1}$ and 15,000 min$^{-1}$ when measured in situ. Machine vibration diagnostics at early operational stages provide information of potential malfunctions, which can help to avoid unexpected repairs and possible breakage of the entire unit. Scheduled periodic vibration measurements could ensure satisfactory operation of the machine. The spectrum analyzer used in the study provided the means for necessary diagnostic measurements of the rotating machinery units, and can be used for machine vibration diagnostics, bearing diagnostics, lubrication evaluation diagnostics, planning of machine maintenance, and prevention of machine downtime during the season.

The vibration instrumentation included the vibration spectrum analyzer Adash A4900 Vibrio M, accelerometer type AC 150, with a magnetic base, with a sensitivity of 100 mV/g and an accuracy range of ±2.5% [19]. The Adash vibration analyzer uses DDS software [19,20], which allows templates of the vibration results to be created. These templates can be chosen from those included in the software, or new templates can be constructed which reflect the characteristics of the specific tested equipment. The DDS software provides 17 separate data graphs of the tested unit. Depending on the specific requirements of the mechanical unit, only the graphs which best characterize the tested equipment can be chosen. This helps to quickly and reliably determine the bearing deterioration status, and to identify major defects. The most common evaluation of bearing condition is called Fault Source Identification Tool (FASIT) (Adash spol. s.r.o., Ostrava, Czech Republic), and was appraised in this study. The graphs displayed the total vibration severity of the machine, bearing condition, speed (value only, no severity indication), severity of imbalance, severity of looseness, severity of misalignment, and severity of other sources. Each graph was composed of several alarms coded with colors. The different colors indicated the severity of the bearing condition; green indicated the bearing was in good condition and no defects were found, yellow was an alert status and suggested the initiation of some defects (the machine could still be operated, but more detailed inspection and possible repairs should be scheduled), and red showed that serious defects existed, and the operation of the equipment should be stopped. Continuous operation at this status could lead to a catastrophic failure and expensive repairs [21].

The locations of the accelerometer placement on the bearing housing of the threshing drum are shown in Figure 1. The vibration measurements were obtained for 600 min$^{-1}$, 900 min$^{-1}$, and 1180 min$^{-1}$ threshing drum rotation speeds, used for wheat crop harvesting. The bearing vibrations were measured in three perpendicular radial directions [21] to determine which location provides the most pertinent information on the bearing condition.

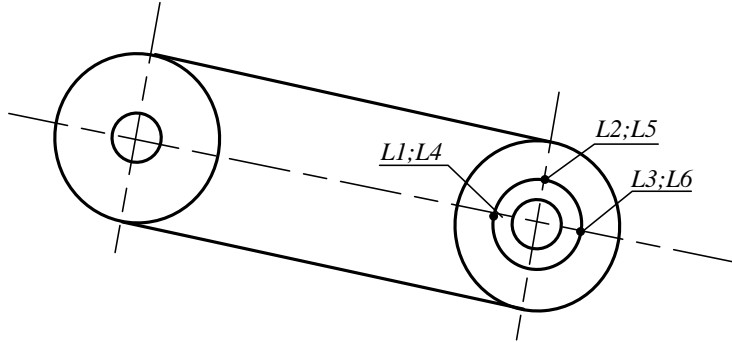

**Figure 1.** Bearing vibration measuring points: L1, L2, L3; accelerometer placed directly on a bearing seat. L4, L5, L6; accelerometer placed on a manufactured bracket.

There are many different bearing housings of the combine harvester shaft. The bearing housing can be made of cast iron or steel, with edges large enough to attach accelerometers. However, in cases where the bearing seats are made from a relatively thin metal plate, there is not enough room for accelerometer installation. In these cases, a metal bracket can be manufactured and installed at the bearing seating, to which an accelerometer is attached. In the present investigation, a bracket was constructed and installed, as shown in Figure 2a. Bracket dimensions presented in mm.

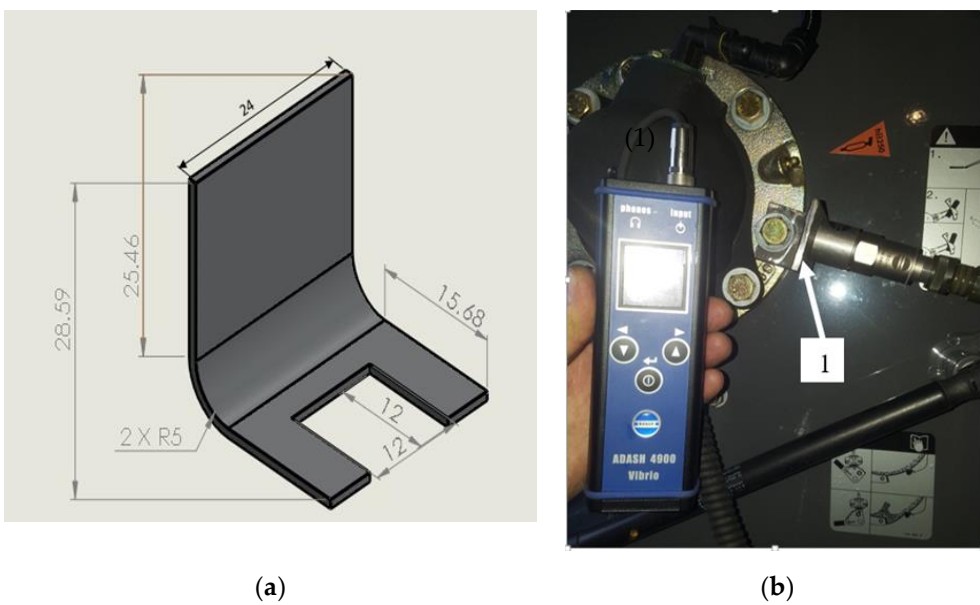

(**a**)　　　　　　　　　　　　　　　　　　(**b**)

**Figure 2.** Manufactured bracket for vibration measuring points that are difficult to access: (**a**) dimension of the bracket; (**b**) accelerometer placed on the bracket (1).

The surface where the accelerometer or the bracket is attached should be clean, and preferably unpainted. In addition, the location should be flat to ensure proper bonding, and the best conditions selected so that the accelerometer measures vibrations in the direction that is normal for the shaft.

The bracket was bolted onto the housing of the bearing. After loosening the bolts of the bearing holders, the bracket was inserted between the bearing holder and the mounting

bolt and tightened back, to ensure a reliable and stationary connection and proper vibration measurement (Figure 2b). The amount of energy loss for a bolted joint depends on the material, surface finish, contact area, bolt tension, lubrication, and fretting corrosion.

Modal analysis and harmonic response analysis were performed for the bracket and housing bearing in ANSYS software to identify and eradicate the common natural frequencies which could result in resonance.

To take a genuine vibration measurement of the bearing from the added manufactured materials, it is necessary to develop a suitable design, with appropriate material selection and simulation of that material to avoid the resonance phenomenon. Before conducting vibration measurement using the added bracket, the selection of material was discussed from a material properties point of view. The bracket design aimed to avoid vibration lapse, and modal analysis was undertaken to avoid resonance due to similar natural frequencies in the associated parts. The bearing, bracket, and housing design were molded in SolidWorks, as shown in Figure 3. The material properties of the components used in the modeling are shown in Table 1. The dimensions of the bearing housing and bearing are shown in Figures A1 and A2 respectively.

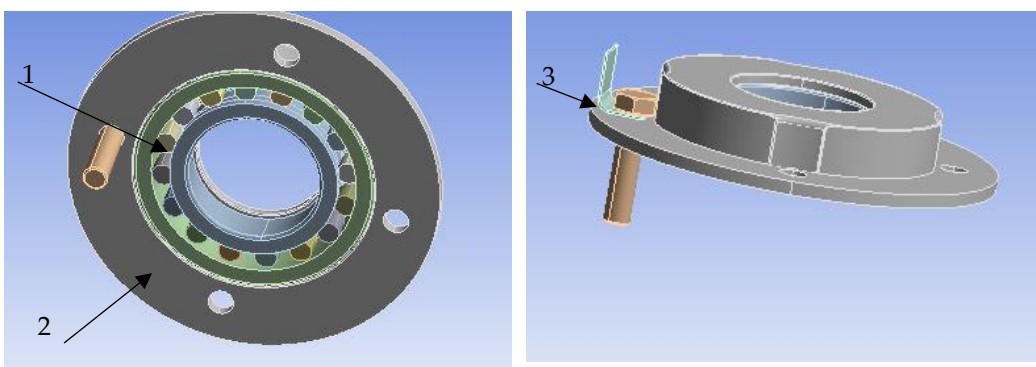

**Figure 3.** CAD model: (**1**) bearing type 20213kTVC3; (**2**) bearing housing; (**3**) bracket.

**Table 1.** Material properties.

| Properties | Steel 45 | Stainless Steel | Gray Cast Iron |
|---|---|---|---|
| Density (kg/m$^3$) | 7800 | 7850 | 7300 |
| Young's modulus (Pa) | $2.1 \times 10^{11}$ | $2 \times 10^{11}$ | $16 \times 10^{11}$ |
| Poisson's ratio | 0.3 | 0.3 | 0.255 |
| Ultimate tensile strength (Pa) | $6.86 \times 10^8$ | $5.5 \times 10^8$ | $2.76 \times 10^8$ |
| Tensile yield strength (Pa) | $4.9 \times 10^8$ | $2.5 \times 10^8$ | $1.03 \times 10^9$ |
| Bulk modulus (Pa) | $1.63 \times 10^{11}$ | $1.6667 \times 10^{11}$ | $5.4 \times 10^{10}$ |
| Shear modulus (Pa) | $8 \times 10^{10}$ | $7.69 \times 10^{11}$ | $6.55 \times 10^{10}$ |

The load for the bearing was identified from the threshing components (flywheel, housing, shaft, threshing drum, chain and sprocket drive, and blower) mounted on the shaft. The modal analysis was performed for the system components consisting of the bearing, housing, bolt, and bracket. The frictional connection was defined with a frictional coefficient between the bearing races and housing (0.4), rollers and bearing races (0.21), and bolts and brackets (0.23).

The radial load was applied as the bearing load on the inner race of the bearing in the system, and from the harmonic response analysis system oscillation at its natural frequency due to forced vibration of the bracket, maximum displacement was obtained. The natural frequencies (eigenvalues) of the bracket obtained from the modal analysis were compared

to the natural frequencies of the bearing housing, bearing, and bolt to analyze the feasibility of the design and material of the bracket.

## 3. Results and Discussion

### 3.1. Bracket Selection

Quenched Steel 45 was selected as the material of the bracket due to its high stiffness. Its physical properties were: Young's modulus = 210 GPa, Poisson's ratio = 0.3, shear modulus = 80 GPa, and density = 7800 kg/m$^3$ [22]. The modal analysis of the housing was performed in ANSYS for bracket dimensions of $80 \times 24 \times 1$ mm, and the extrude was introduced as $12 \times 12$ mm. The mesh statistics are shown in Table 2.

**Table 2.** Mesh statistics.

|  | **Bracket** | **Components (Bearing, Housing, Bracket, and Bolt)** |
|---|---|---|
| Nodes | 4831 | 57,284 |
| Element | 2182 | 21,317 |

The meshed system is shown in Figure 4.

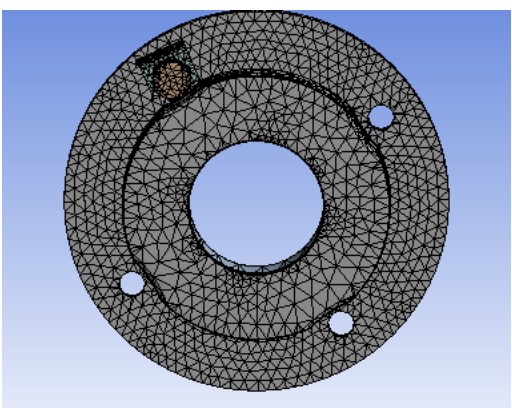

**Figure 4.** The meshed system.

The modal analysis of the bracket was performed in ANSYS, and the eigenvalues were 925.98 Hz, 1995.8 Hz, 3681.8 Hz, 4156 Hz, 9412.4 Hz, and 11,828 Hz (Figure 5). These are the first five-mode shapes of the bracket. For modal analysis, the bracket is fixed on the base and at the contact surface region with a bolt. Furthermore, the modal analysis was performed for the system components consisting of the bearing, housing, bolt, and bracket. The frictional connection was defined between the bearing races and housing (0.4), rollers and bearing races (0.21), and bolts and bracket (0.23). The active mode shape for the bracket in the system was 1363.58 Hz, as shown in Figure 6.

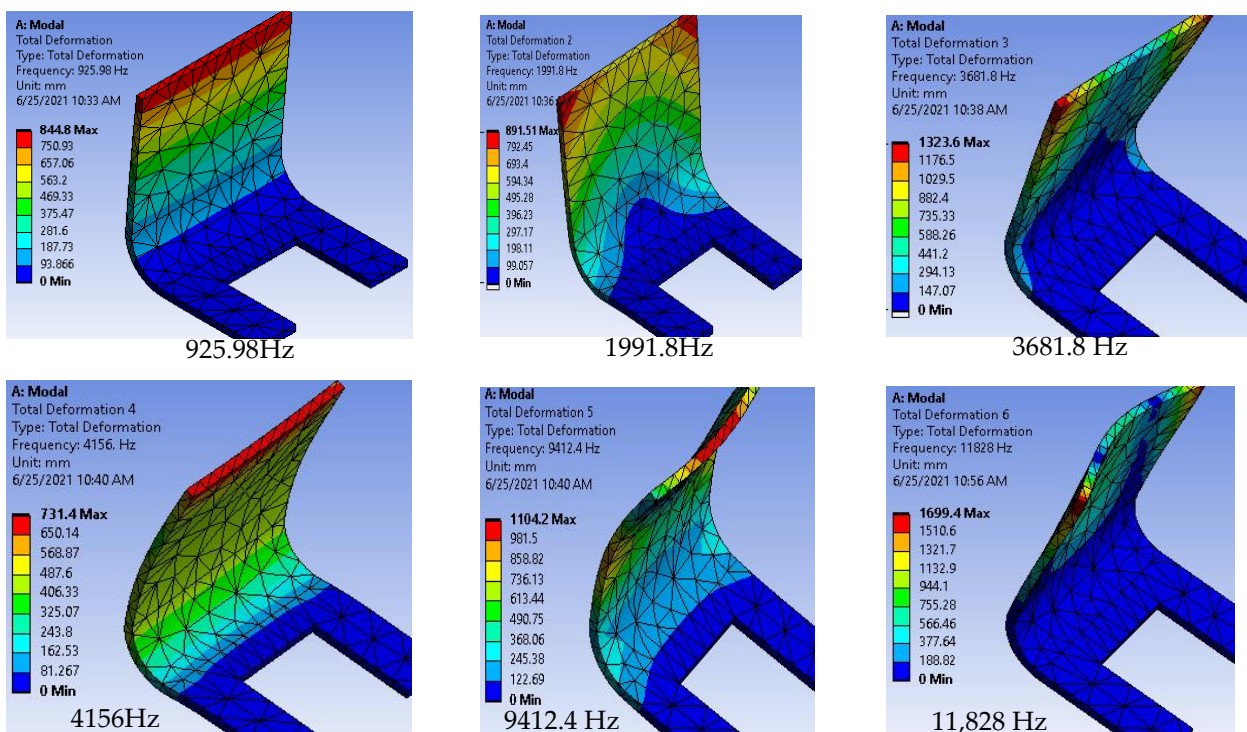

**Figure 5.** Eigenvalues and eigenvector of the bracket.

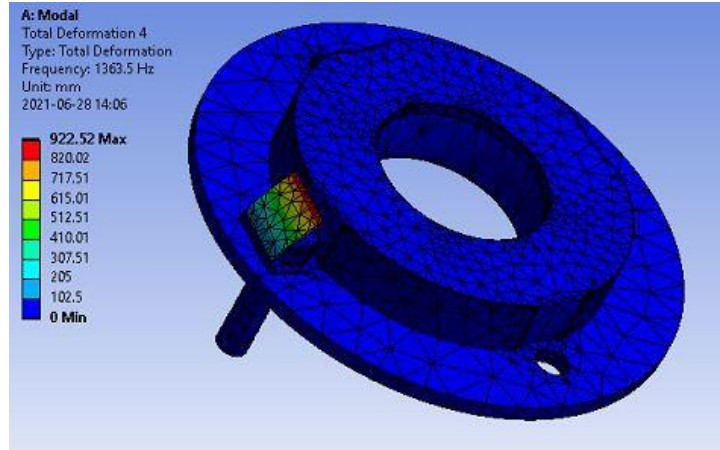

**Figure 6.** Mode shape of the bracket in the system modal analysis.

The radial load applied was obtained from the sum of the components: flywheel (392 N), drum (196 N), shaft (7.84 N), blower/fan (39.2 N), and the pull force from the belt drive (11.637 N). Belt pull is the force which is employed on the shaft which relies on drive power, rotation speed driven, pitch diameter of the driving pulley and driving factor. The pull force from belt drive can be calculated as [23]:

$$\text{Belt Pull} = (126000 \times \text{HP} \times \text{f})/(\text{RPM} \times \text{PD}) = 2.615 \text{ lbs.} = 11.637 \text{ N} \tag{1}$$

where HP is the drive power to the threshing (16.5 KW), f is the driving factor based on the synchronous belt drive (1.3) [24], RPM is the rotation speed of the small sprocket (1180 $\text{min}^{-1}$), and PD is the pitch diameter of the small sprocket (29.97 mm).

The total radial load on the bearing was calculated as 646.677 N. The bearing load is applied to the inner race of the bearing. The motion pattern of a system oscillation at its natural frequency by forced vibration in harmonic response analysis for the bracket

maximum displacement was found at 936 Hz of 9.322 mm displacement. Thus, frequencies near 936 Hz need to be avoided for other components to avoid resonance in the system. Figure 7 represents the displacement of a bracket in forced vibration using harmonic response analysis.

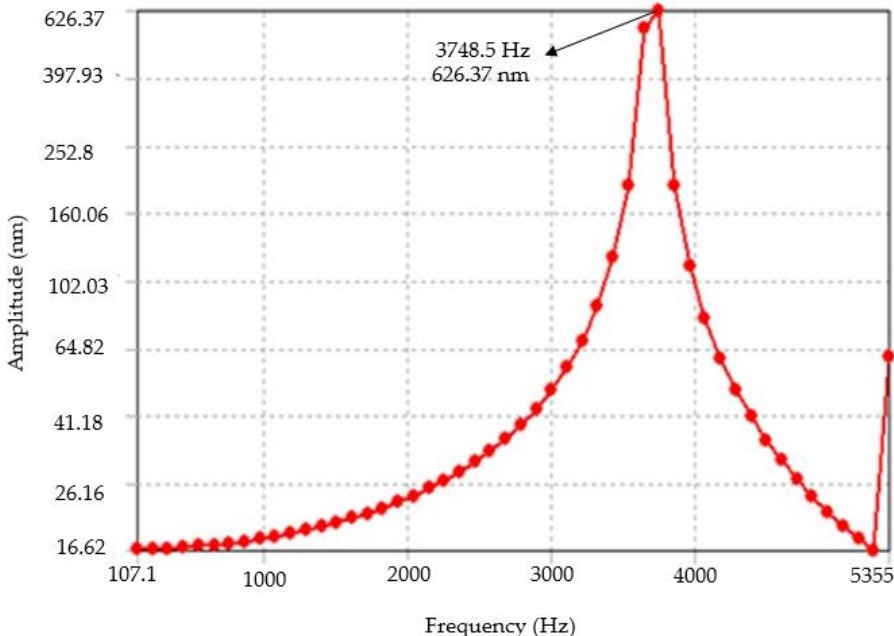

**Figure 7.** Harmonic response of the bearing.

Similarly, from the modal and harmonic response analysis of bearing housing, the maximum displacement of 0.00062 mm was obtained at 3748.5 Hz (Figure 7) for the radial load applied to the housing. In addition, bearing defect frequencies were calculated [3,25]. The calculated bearing defect frequency of the outer ring (f-RPI) was 118 Hz, the inner ring (f-RPV) was 175 Hz, and the rollers defect (f-RDD) was 45 Hz. Since the eigenvalues of the associated components vary, the steel 45 material and the dimensions were considered for the manufacture of the bracket for vibration analysis.

*3.2. Vibration Analysis*

To determine the bearing vibration differences with and without a bracket, the easily accessible bearing housing of the combine harvester was selected. The measurements were performed comparing the data. The main objective of these measurements was to develop a database to determine the differences between cases with the manufactured bracket and without the bracket. These results are presented in Figure 8. At some locations where the manufactured bracket could be properly fitted to the surface. The RMS data are mostly used for bearing vibration evaluation. The measurement difference was calculated from the obtained results, and the given coefficient was used for evaluation of the threshing drum bearings. The collected data showed 17.92% increased vibration while using the bracket (Figure 8). Thus, if the metal bracket could be properly installed at some locations, the bracket can be used to assist in vibration measurements of the threshing drum bearings.

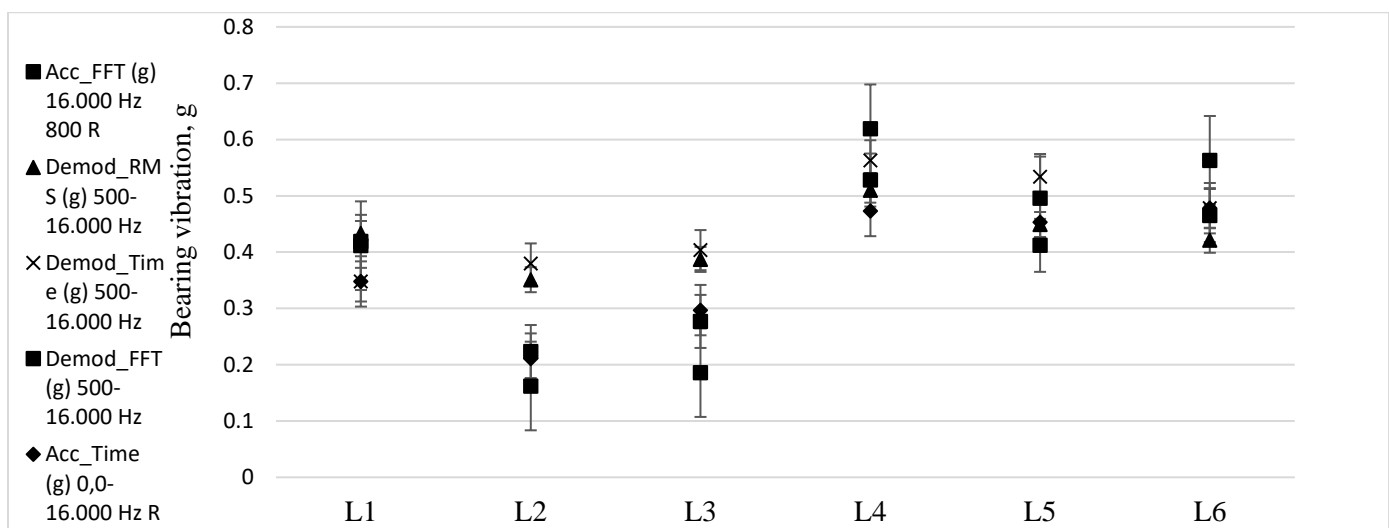

**Figure 8.** Bearing vibration acceleration at 3050 min⁻¹ speed. L1, L2, L3; accelerometer placed directly on a bearing seat. L4, L5, L6; accelerometer placed on a manufactured bracket.

The vibrations at the right side of the threshing drum bearings were measured and transferred to a personal computer. TheFACIT-type graphs were obtained, from which the bearing status could be determined. For the bearing status determination the acceleration by fast Fourier transformation (Acc_FFT), root mean square demodulation (Demod_RMS), time–frequency demodulation (Demod_Time), fast Fourier transform demodulation (Demod_FFT), and acceleration time (Acc_Time) graph were selected. The FACIT sub-figures, provided by the ADASH vibrometer, show the extended and more detailed bearing vibration status: the bearing condition; speed (value only, no severity indication); unbalance; looseness; misalignment. As was mentioned above, each graph is composed of several alarms, coded with colors. If all values are in the green color area, it means good condition and no defects of the bearing. After performing the analysis using the vibration response spectrum, no significant changes in bearing defect frequencies were observed. This was expected, since the combined threshing drum bearings were relatively new.

From the measured vibration results, the most important vibration parameters were selected and further analyzed with Excel 2010 software. A summary of these results is given in Figures 9–11. Then, from the analysis of the vibration data, it was possible to determine key parameters, which most accurately identified possible bearing defects. According to various investigators, analysis using the mean square vibration velocity in the frequency range of 10–1000 Hz cannot be used as the key parameter in the evaluation of the technical condition of slow-rotating bearings. Vibration parameters such as Vrms, g (5–16 kHz), and gr (0.5–16 kHz) do not properly describe the condition of the bearing. The most appropriate assessment of bearing conditions is by FFT spectrum analysis [26,27]. To determine the time before the bearing starts to show the first signs of wear, periodic monitoring of bearing vibrations is needed. The results from the right side of the threshing drum bearing were transferred from the Adash 4900 Vibrio vibration analyzer to a personal computer. The results in the FASIT-type graph represent the bearing vibrations data.

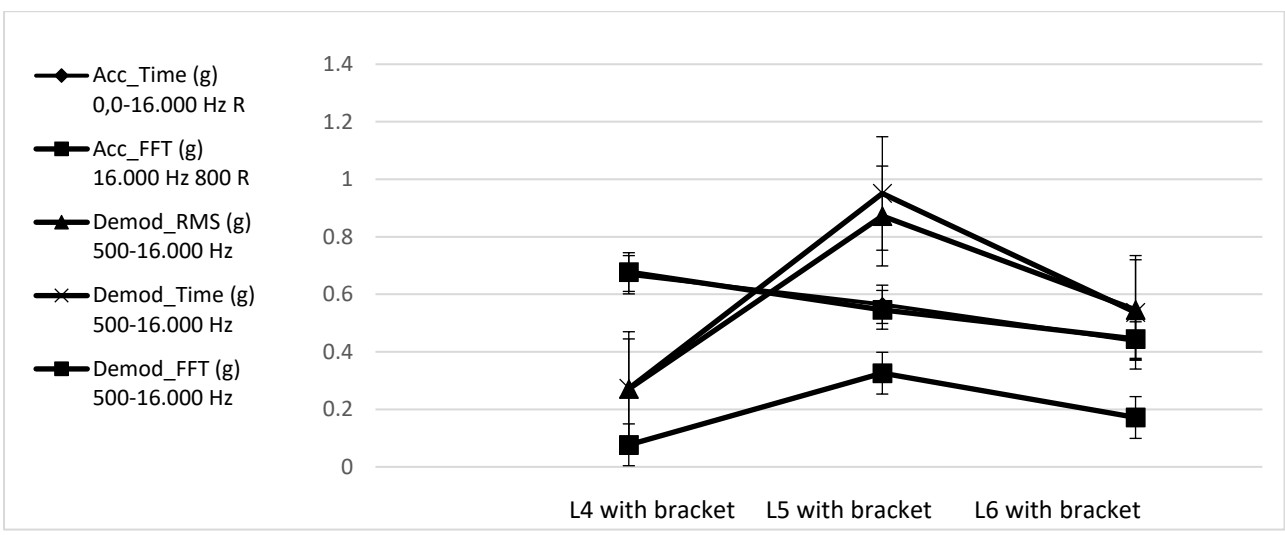

**Figure 9.** Bearing vibration acceleration at a speed of 600 min⁻¹.

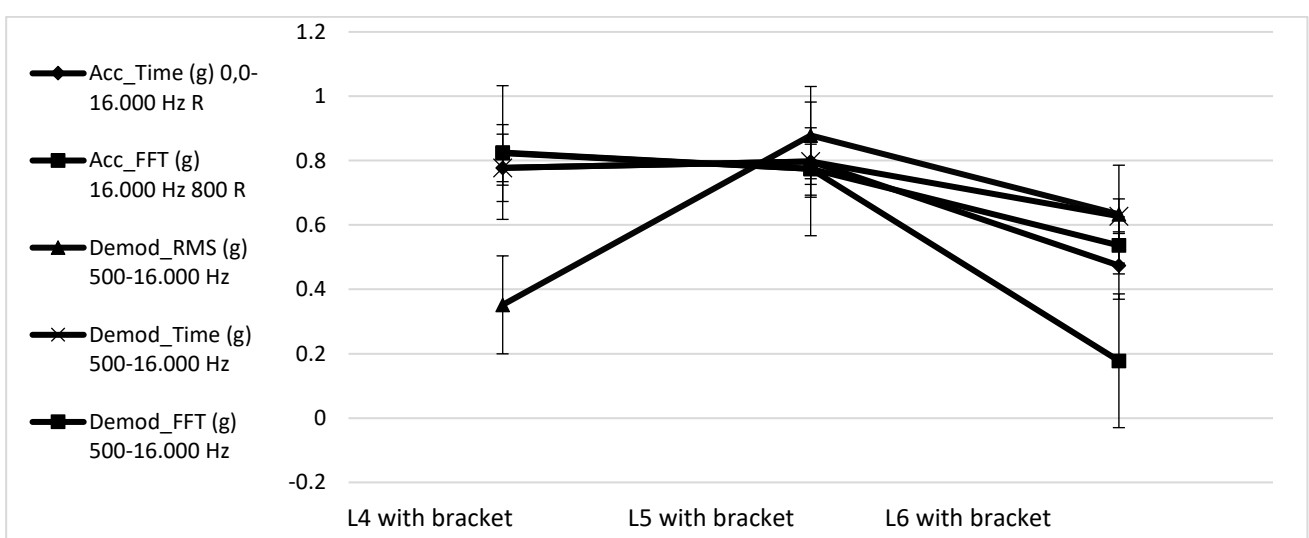

**Figure 10.** Bearing vibration acceleration at a speed of 900 min⁻¹.

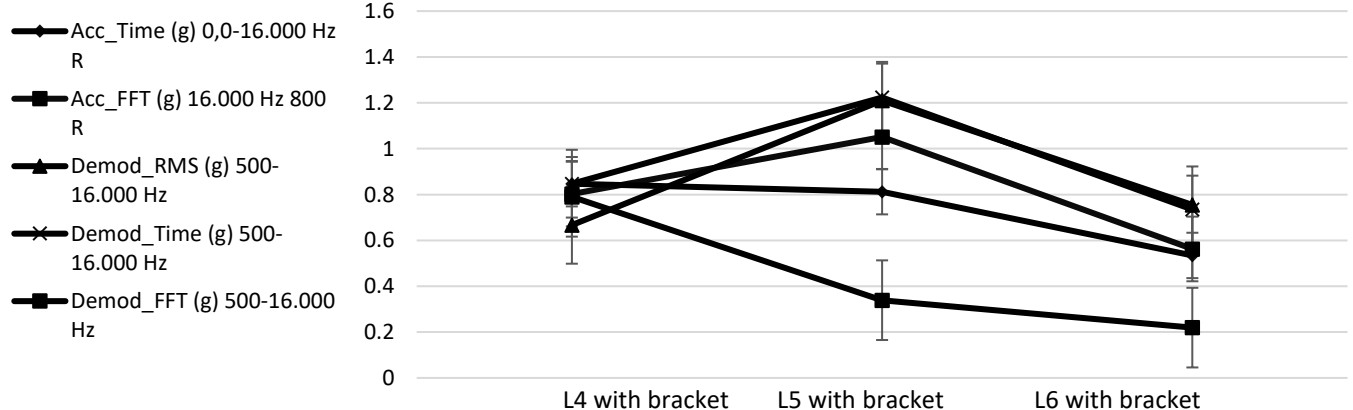

**Figure 11.** Bearing vibration acceleration at a speed of 1180 min⁻¹.

The obtained results indicate that the largest vibrations were at the L5 location, at a rotation speed of 1180 min⁻¹. This indicates that the largest vibrations of the bearings

were in the vertical direction. The lowest vibrations were measured at locations L4 and L6. After vibration data analysis, it was concluded that this bearing had no defects. From the obtained results and data analysis, it can be stated that the average values of the vibration velocity (in the frequency interval of 10–1000 Hz), in combination with other measured parameters, can be used to assess the combine threshing drum's bearing condition.

For the study, combine threshing drum bearings with an output of 3590 MH were used. Measurements were taken using the same methodology as for testing new bearings. After performing the analysis using the vibration response spectrum, significant changes in bearing defect frequencies (25 Hz, 101 Hz) were observed (Figure 12). After comparing the theoretically calculated defect frequencies (the defect frequency of the outer ring (f-RPI) was 118 Hz, the inner ring (f-RPV) was 175 Hz, and the rollers defect (f-RDD) was 45 Hz.), it can be stated that the studied bearing vibrations were caused by a defect of the outer bearing ring.

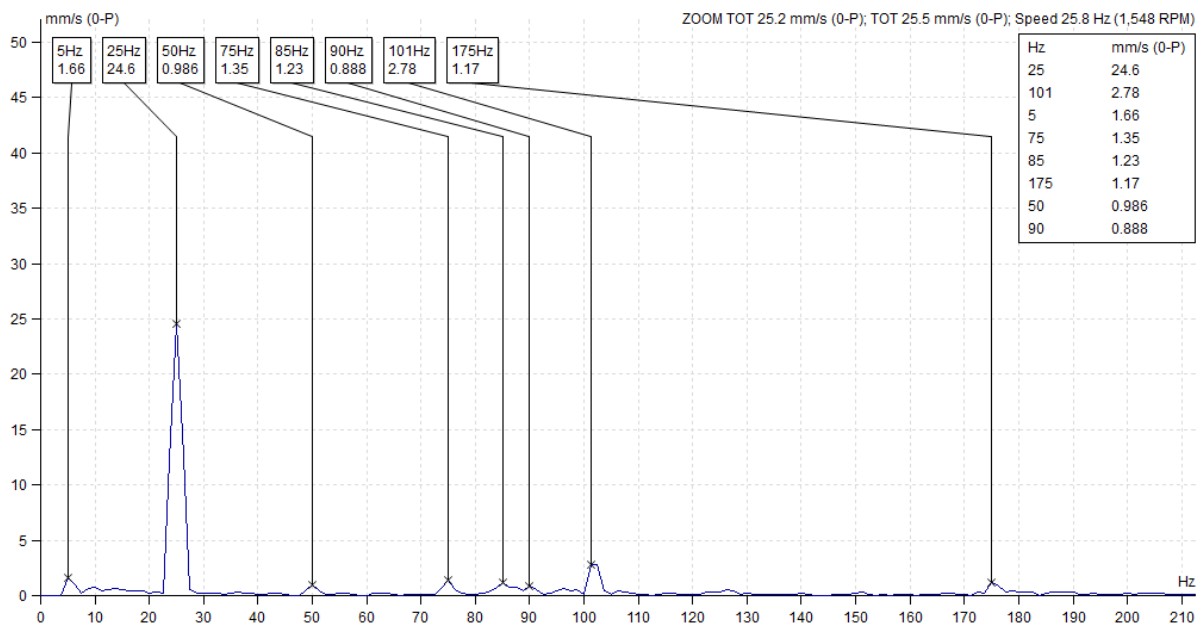

**Figure 12.** Bearing vibration spectrum.

When analyzing the RMS values, these values should be as close to zero as possible. Figure 13 shows that the RMS values of the assessed bearings in the range up to 6000 Hz reached 0.188 g (RMS), which indicates insufficient bearing stability.

To prevent threshing drum failure in the combine, it is recommended to replace the bearing with a new one, or to have it additionally checked by another diagnostic method. A microscope was selected to check the condition of the used bearing again.

Before disassembly of the threshing drum bearing, the vibrations produced by the bearing were measured in a running combine. In addition, a smart 5MP PRO digital microscope was used to identify existing bearing defects after bearing dismounting. The resulting visual material is presented in Figure 14. The image of the used bearing (Figure 14a) clearly shows the surface damage. The damage is presented by cross lines in the inner ring. The inner ring defect determined by the vibration analysis method has been proved by microscope image (Figure 14a). Comparing the microscope image of the new bearing (Figure 14b), the surface of the inner ring is smooth and of good quality.

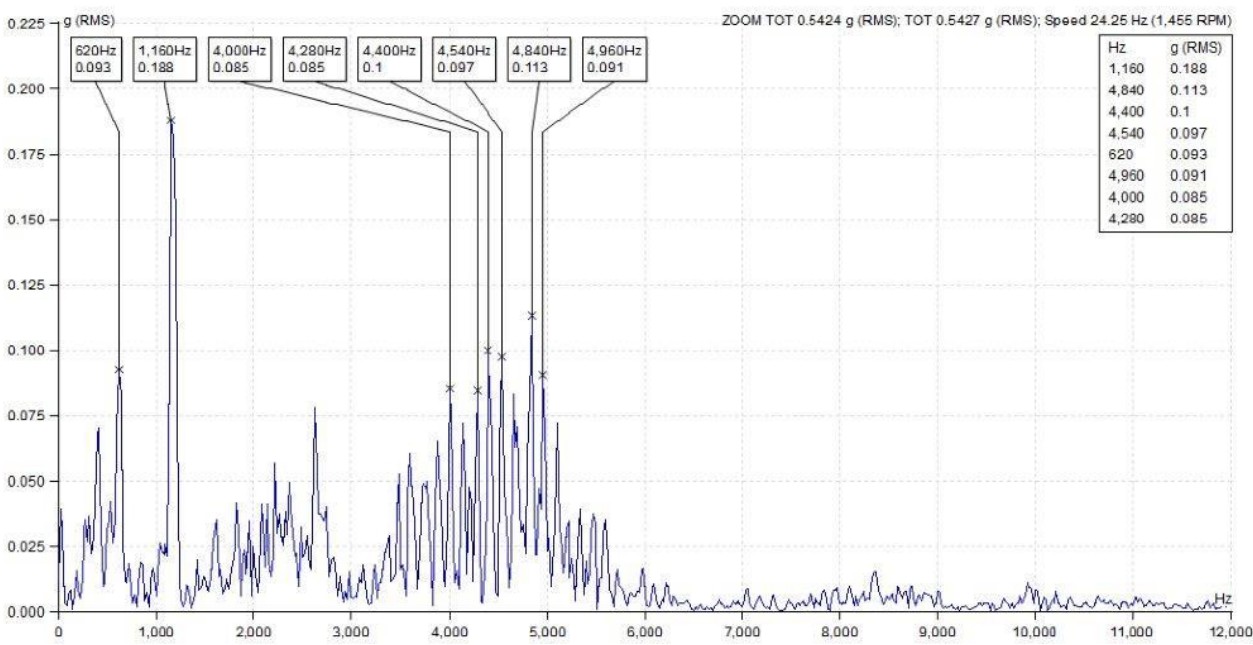

**Figure 13.** Bearing vibration spectrum in the range of 0–12,000 Hz.

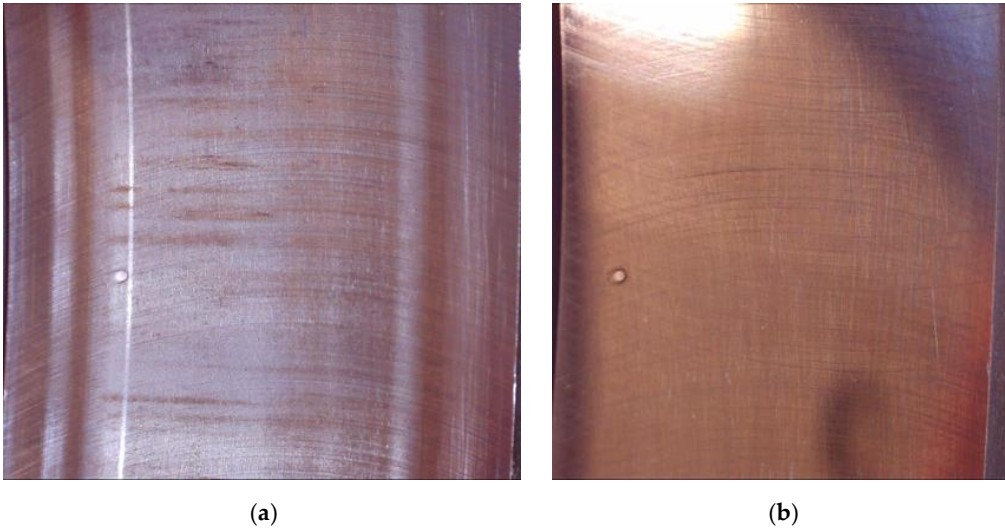

(**a**)                                                                    (**b**)

**Figure 14.** Bearing surface view of the inner ring. (**a**) View of the used bearing inner ring; (**b**) view of the new bearing inner ring.

In summary, utilizing the bracket developed in this paper, vibration instrumentation could be used for diagnostics of the bearing conditions of agriculture machines, without dismounting the bearing. If the measurement values are in green, the bearing is still suitable for use. If the value of the bearing is in red, they are at the limit values of vibrations. In this case, the bearing must be changed. The investigated threshing drum bearing with an output of 3590 MH possibly could be used for a significant period of time. Therefore, even small defects in the bearing could cause a breakdown of the combine harvester requiring several hours to repair. During this breakdown time, a typical combine harvester could harvest from 24–30 ha [28]. Therefore, changing the bearing before a breakdown occurs can reduce unexpected expenses. Timely replacement of the bearings ensures smooth combine harvester operation during the season. At the same time, bearing vibration tests make it easier and simpler to evaluate the bearing condition of combine harvesters and other agricultural machines under standard operating conditions. Further, the change of

vibration energy exerted could be used for delamination defect analysis of the bracket in future investigations [29].

## 4. Conclusions

To evaluate the threshing drum technical conditions of a combine, and to identify defects while the combine is operating in the field, the most appropriate vibration signals were chosen. FFT spectrum analysis provides the required vibration information, and is best suited for the assessment of the bearing quality condition.

The initial vibration measurements at the selected locations of the bearing seat (L1, L2, L3) indicated some differences in their values. It was determined that bearing vibrations could be best characterized at the bearing seat (L2) and at the manufactured bracket (L5), where the largest power force on the belt pulleys is acting in the vertical direction. The manufactured bracket could be used to assist in obtaining vibration data at locations where attaching an accelerometer directly is difficult. Based on these results, the technical condition of a combine's threshing drum bearings can be adequately assessed.

To prevent the combine from suffering threshing drum failure, it is recommended to replace bearings or to have them additionally checked by another diagnostic method regularly. The application of such a bearing condition assessment methodology could support a safer and smoother operation of the threshing apparatus, and the selection of the most appropriate technological parameters during harvesting.

**Author Contributions:** Conceptualization, E.J. and A.J.; methodology, E.J., A.J. and S.B.; software, S.B.; validation, E.J. and A.J.; formal analysis, E.J., A.J. and S.B.; investigation, E.J., A.J. and S.B.; resources, E.J. and A.J.; data curation, E.J., A.J. and S.B., writing—original draft preparation, E.J., A.J. and S.B.; writing—review and editing, E.J. and A.J.; visualization, E.J., A.J. and S.B.; supervision, E.J.; project administration, E.J.; funding acquisition, E.J., A.J. and S.B. All authors have read and agreed to the published version of the manuscript.

**Funding:** This research received no external funding.

**Institutional Review Board Statement:** Not applicable.

**Informed Consent Statement:** Informed consent was obtained from all subjects involved in the study.

**Acknowledgments:** The authors would like to thank the Institute of Agricultural Engineering and Safety of the Agriculture Academy of Vytautas Magnus University for the test development and support.

**Conflicts of Interest:** The authors declare no conflict of interest.

## Appendix A

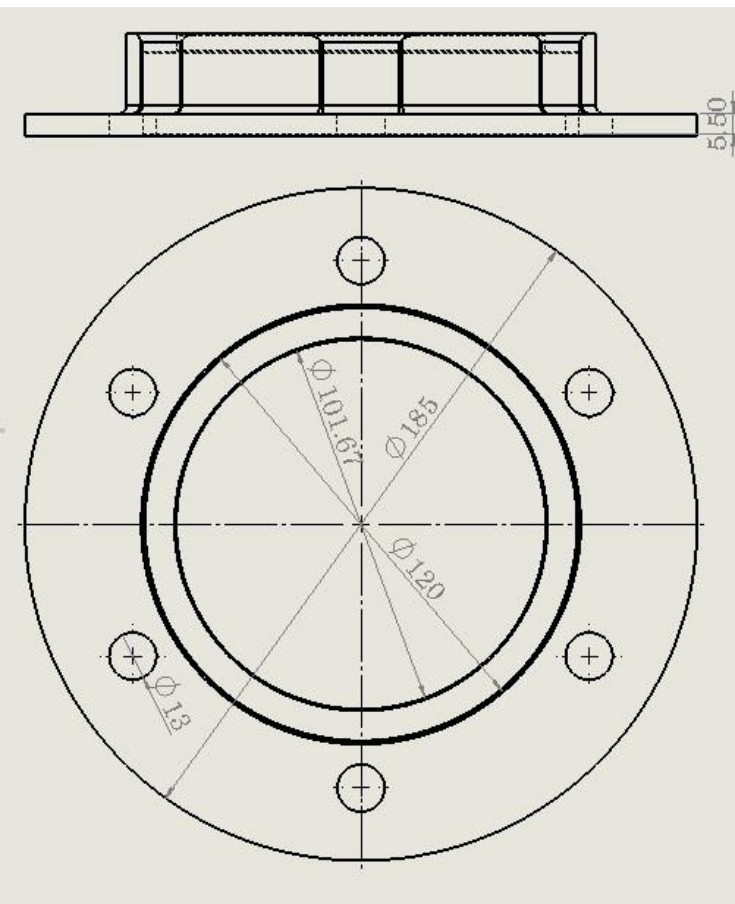

**Figure A1.** Bearing housing dimensions.

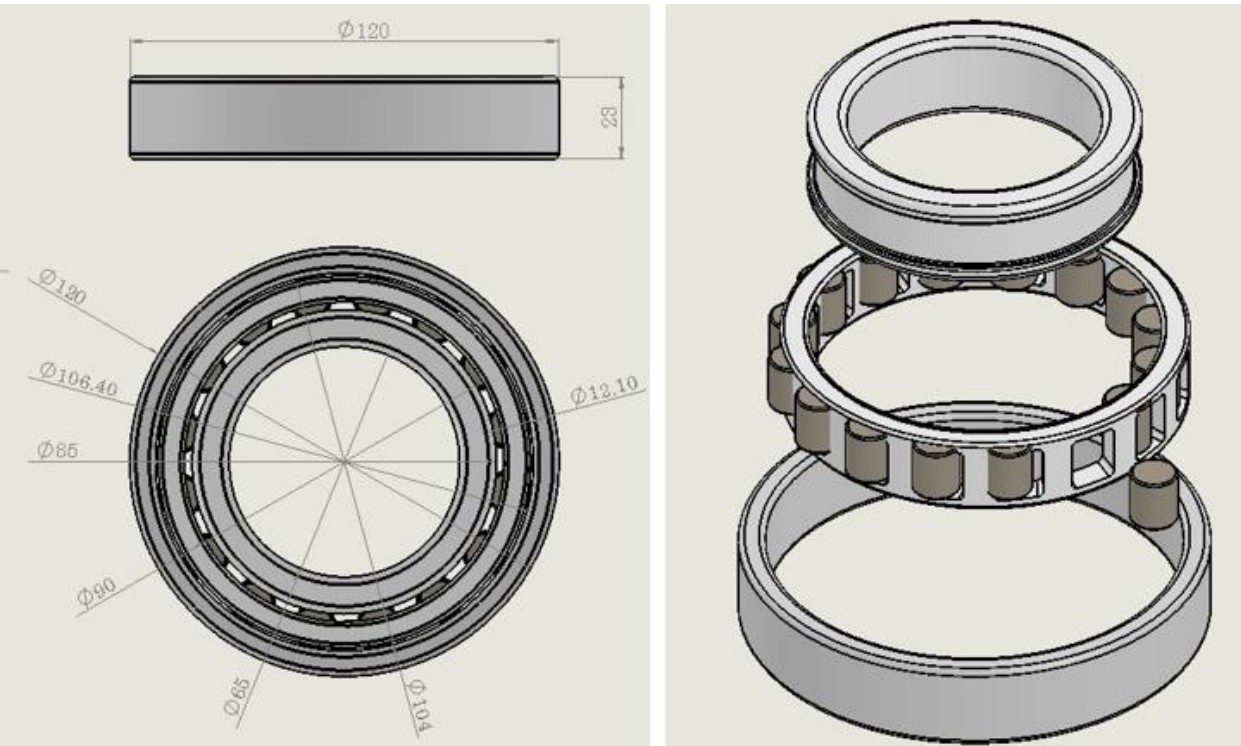

**Figure A2.** 20213 k TVP C3 bearing dimensions.

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
