# Peer review of "Proper Technical Maintenance of Combine Harvester Rolling Bearings for Smooth and Continuous Performance for Grain Crop Agrotechnical Requirements"

_applsci, doi:10.3390/app11188605_

Round 1

Reviewer 1 Report

  1. Fig.8 and Fig.10-12, the bar chart should be changed into the line chart, bar charts always used in management discipline
  2. Fig.9 should be removed, the figure quality is not good enough. Can you rebulid the sub-figures using the collected information to show the important information to  readers.
  3. It is inevitable to cause the surface damage of the inner ring during the field operation, they are still can be used with such condition. How to judge the work condition of the bearing using the mentioned method  in the paper, in which condtion they should change a new one.

Author Response

We would like to thank you for the very great comments of the reviewer, as they help us to improve the manuscript and to inform you that our manuscript was revised after considering all the comments of the reviewers.

Changes were made in the manuscript by using green color and track changes.

  1.  Fig. 8 and Fig. 10-12, the bar chart should be changed into the line chart, bar charts are always used in the management discipline.

Response: We agree with the comment and corrected Fig. 8 and Fig. 10-12 (the bar chart changed into the line chart).

  1. Fig. 9 should be removed, the figure quality is not good enough. Can you rebuild the sub-figures using the collected information to show the important information to readers.

Response: We agree with the comment and we rebuild Fig. 9 into the sub-figures.

  1. It is inevitable to cause surface damage of the inner ring during the field operation, they are still can be used with such conditions. How to judge the work condition of the bearing using the mentioned method in the paper, in which condition they should change a new one.

Response:  We accept the comment and agree that bearing possibly could be used for some period. Therefore, after observation, even small defects of the bearing could cause a breakdown of the combine harvester for 4 to 5 hours of repair. During this breakdown time, combine harvester could harvest from 24 to 30 ha. And, at a yield of 6 t ha-1 with a grain price of 200 Eur t-1, it can cause an expense close to 36 000 Eur [29]. Therefore, the suggestion is to change the bearing before the break to control the unexpected expenses.

Reviewer 2 Report

1. The authors use modal analysis to show the rationality of selecting the Quenched Steel 45, but there is a lack of comparison with other materials to present the necessity of selecting the Quenched Steel 45.

2. In Fig. 8, the authors should clearly indicate the data with and without bracket, and analyze the results to illustrate the necessity of using the proposed bracket.

3. According to various investigators, the authors select the FFT as the analysis method. The authors should give references to justify these claims.

Author Response

We would like to thank you for the very great comments of the reviewer, as they help us to improve the manuscript and to inform you that our manuscript was revised after considering all the comments of the reviewers.

Changes were made in the manuscript by using yellow color and track changes.

  1. In Fig. 8, the authors should clearly indicate the data with and without bracket and analyze the results to illustrate the necessity of using the proposed bracket.

Response:  Thank you for the comment. In Fig.8 the data with and without a bracket were more clarified. The bearing holders in combine are different in a size. Some bearing holders are cast with a large area for mounting the accelerometer, and some holders are very thin made of a 5 mm thick sheet of metal. For such a type of bearing holder, the manufactured bracket for the accelerometer is needed. A bracket was made to have a possibility to measure bearing vibrations for all types of bearing holders. Otherwise, the bearing must be dismounted for a check. After dismounting the threshing drum bearings, the installation of the old bearing is not reasonable.

  1. According to various investigators, the authors select the FFT as the analysis method. The authors should give references to justify these claims.

Response:  Thank you for the comment. The selected method was justified by references [27, 28].

Reviewer 3 Report

In this paper, E. Jotautiene et al. carried out a study regarding the technical maintenance of a combine harvester. Unfortunately, I can not recommend this paper for publication in the journal of Applied Sciences. My concerns are as follow: 

Point 1: Which is the main novelty of the paper? The authors should explain what it is the major contribution of this paper to the literature.

Point 2: The criteria for the location of the measurement positions should be introduced in the manuscript. Why do you choose these locations?

Point 3: It would highly recommend to include further information about the Modal Analysis via ANSYS software (element type, material properties, dimensions, etc.)

Point 4: Apart from that, I would like to know if the results from the numerical simulations have been verified and validated?

Point 5: As a general comment, the results from the figures should be discussed in more detail through the manuscript.

Author Response

We would like to thank you for the very great comments of the reviewer, as they help us to improve the manuscript and to inform you that our manuscript was revised after considering all the comments of the reviewers.

Changes were made in the manuscript by using turquoise color and track changes.

  1. Which is the main novelty of the paper? The authors should explain what it is the major contribution of this paper to the literature?

Response:  Thank you for the comment. The main novelty of the article is the presence of the bracket and its implementation for the bearing test at the hard accessible places of mobile agricultural machines. Vibration analyzer Adash 4900 Vibrio is dedicated to measure the bearing vibrations for industrial stationary machines like many others. Therefore, the problem comes with how to suit the existing bearing testing methodology for mobile agricultural machines. By presenting and implementing the bracket, the testing equipment could be used for measuring the bearing vibrations for agriculture machines too, without dismounting the bearing. After dismounting the threshing drum bearings, the installation of the old bearing is not reasonable.

  1. The criteria for the location of the measurement positions should be introduced in the manuscript. Why do you choose these locations?

Response: The measurement points must be selected as close to the bearing as possible. In other researchers’ methodology, the accelerometer was mounted in an axial as well as in the radial running direction of the bearing [7, 26]. Standard ISO 10861-3 recommends that accelerometers must be mounted in such a way that the resulting direction of oscillation is related to the applied dynamic [18]. Two points arranged in a perpendicular circle are usually required. An accelerometer of the measuring device, for assessing the condition of the bearings, mounted on the bearing points in the radial direction of rotation of the shaft.  As shown in Fig.1, taking into account the different loads, three measuring points were selected.

  1. It would highly recommend to include further information about the Modal Analysis via ANSYS software (element type, material properties, dimensions, etc.)

Response: The bearing 20213 K TVP C3 modal is imported from the CAD library and material is assigned as structural steel to the races and rollers and PV6 plastic for the cage. The bearing dimension of the bearing is standard. The material of housing is identified as grey cast iron. The dimension is obtained from Agrodoctor website and modified from the part measurement for designing. The bracket material and dimension are well discussed in the paper. The M4 stainless steel bolt used also has a standard dimension. The material properties are added to the manuscript. The design description of bearing and housing is added to Appendix A.

  1. Apart from that, I would like to know if the results from the numerical simulations have been verified and validated?

     Response: Modal analysis of the simulation model being validated resemblance the results of other (valid) homogeneous models [16, 17]. For harmonic response analysis for the same modal, the radial loads applied are from the weights of the components (flywheel, bearing, 

Round 2

Reviewer 2 Report

None

Author Response

Thank you for the comment. The English language is submitted for revision with an MDPI language expert. 

Reviewer 3 Report

The manuscript has been improved, however, there are still a couple of comments from my side before acceptance. 

Point 1) The style of the figures should be revised. The numbers and legend of the figures are difficult to read. 

Point 2) The particular frequencies of the bearing vibration spectrum should be detailed in Figures 13 and 14. I would like to know how the changes of these particular frequencies can be used for the identification of defects. 

Point 3) The following reference deal with the detection of delamination defects in composite marine bulkheads using a NDT vibration based method. This reference could be included in the revised version of the manuscript. 

  • Detection and Quantification of Delamination Failures in Marine Composite Bulkheads via Vibration Energy Variations, Sensors 2021, 21, 2843.

Finally, I also encourage the authors to carry out a revision of the English language and style.

Author Response

We would like to thank you for the very great comments of the reviewer, as they help us to improve the manuscript and to inform you that our manuscript was revised after considering all the comments of the reviewers.

Changes were made in the manuscript by using green color and track changes.

  1. The style of the figures should be revised. The numbers and legend of the figures are difficult to read.

Response: We agree with the comment and corrected Fig. 8 and Fig. 10-12 (the bar chart changed into the line chart) more clearly and visibly.

  1. The particular frequencies of the bearing vibration spectrum should be detailed in Figures 13 and 14. I would like to know how the changes of these particular frequencies can be used for the identification of defects.

Response: Thank you for the comment. Figures 13 and 14 represent the bearing vibration spectrum. The theoretical calculations of bearing defect frequencies are done according to formulas used from literature sources [25, 26]  (the outer ring frequency (f-RPI) is 118Hz, the inner ring (f-RPV) is 175Hz and the rollers defect (f-RDD) is 45Hz). Having the vibration spectrum obtained during the experiment we can compare the largest amplitudes of the calculated and experimental frequencies. For this purpose, the spectral amplitude values are denoted in Figures 13 and 14. If the calculated frequency values are close to the experimental one, it is concluded possible bearing defect.

  1. The following reference deal with the detection of delamination defects in composite marine bulkheads using a NDT vibration based method. This reference could be included in the revised version of the manuscript.

Response:  We accept the comment that the provided reference partially compeer with our objective and deals more detailly with the determination of the severity of the delamination defects in composite panels. As the novelty of this paper is to identify the feasibility of using external material (bracket) for vibration analysis in hard-to-access places of mobile agricultural machines without dismantling the cases. The delamination defects analysis can be the further step of the investigation.

Furthermore, the manuscript is submitted for the English language is revision with MDPI language expert.

Round 3

Reviewer 3 Report

I recommend this paper for publication in the journal of applied sciences.

Author Response

Response to Reviewer 3 Comments

We would like to thank you for the very great comments of the reviewer, as they help us to improve the manuscript and to inform you that our manuscript was revised after considering all the comments of the reviewers.

  1. The English language and style are fine/minor spell check required.

Response: We agree with the comment and corrected the manuscript.
